# Application of Multiple-Source Data Fusion for the Discrimination of Two Botanical Origins of Magnolia Officinalis Cortex Based on E-Nose Measurements, E-Tongue Measurements, and Chemical Analysis

**DOI:** 10.3390/molecules27123892

**Published:** 2022-06-17

**Authors:** Wenguang Jing, Xiaoliang Zhao, Minghua Li, Xiaowen Hu, Xianlong Cheng, Shuangcheng Ma, Feng Wei

**Affiliations:** 1National Institutes for Food and Drug Control, Beijing 100050, China; jingwenguang@nifdc.org.cn (W.J.); bjlmmhh@163.com (M.L.); dreadless@126.com (X.H.); cxl@nifdc.org.cn (X.C.); 2Beijing Key Laboratory of Traditional Chinese Medicine Basic Research on Prevention and Treatment of Major Diseases, Experimental Research Center, China Academy of Chinese Medical Sciences, Beijing 100700, China; zhaoxiaoliang1218@aliyun.com

**Keywords:** Magnoliae Officinalis Cortex, multiple-source data fusion, origin, e-tongue, e-nose, chemical analysis, feature selection, multivariate statistical analysis, discriminative model

## Abstract

*Magnolia officinalis* Rehd. et Wils. and *Magnolia officinalis* Rehd. et Wils. var. *biloba* Rehd. et Wils, as the legal botanical origins of Magnoliae Officinalis Cortex, are almost impossible to distinguish according to their appearance traits with respect to medicinal bark. The application of AFLP molecular markers for differentiating the two origins has not yet been successful. In this study, a combination of e-nose measurements, e-tongue measurements, and chemical analyses coupled with multiple-source data fusion was used to differentiate the two origins. Linear discriminant analysis (LDA) and quadratic discriminant analysis (QDA) were applied to compare the discrimination results. It was shown that the e-nose system presented a good discriminant ability with a low classification error for both LDA and QDA compared with e-tongue measurements and chemical analyses. In addition, the discriminating capacity of LDA for low-level fusion with original data, similar to a combined system, was superior or equal to that acquired individually with the three approaches. For mid-level fusion, the combination of different principals extracted by PCA and variables obtained on the basis of PLS-VIP exhibited an analogous discrimination ability for LDA (classification error 0.0%) and was significantly superior to QDA (classification error 1.67–3.33%). As a result, the combined e-nose, e-tongue, and chemical analysis approach proved to be a powerful tool for differentiating the two origins of Magnoliae Officinalis Cortex.

## 1. Introduction

An increasing amount of attention is being given to traditional Chinese medicines (TCMs) by people from around the world due to their long history and excellent curative effects. TCMs are influenced by their natural environment, geographical region, time of harvest, and other factors that contribute to their unique qualities. Multiple botanical origins are a major feature of TCMs and demonstrate their diversity. However, the multiple botanical origins of medicinal materials also create differences in their appearance traits and odor characteristics, as well as quality discrepancies. The quality and efficacy of prescriptions composed of medicinal materials with various origins are attracting increasing amounts of attention. Therefore, determining ways in which the various sources of medicinal materials can be distinguished and their quality discerned is of great importance. The origin of TCMs is conventionally characterized by their appearance, color, and odor, and can be subjectively evaluated on the basis of morphology and human experience. However, a particular challenge regarding origin discrimination is presented owing to the similarity of both species and their chemical composition.

The electronic nose (e-nose) is an intelligent electronic instrument designed to simulate the process and mechanisms of human olfactory recognition. It is suitable for measuring one or more odoriferous substances in complex systems. The intelligent taste analysis system, the electronic tongue (e-tongue), uses artificial lipid film sensor technology similar to the mechanism of gustatory cells on the human tongue. It can objectively and digitally evaluate the basic flavors of bitterness, astringency, sourness, saltiness, sweetness, and umami for food or medicinal samples. The e-nose and e-tongue fusion system, with the significant advantage of obtaining comprehensive taste and olfactory information, could provide an opportunity to progress from subjective to objective evaluation. Although e-tongue and e-nose systems were initially and predominantly implemented in food flavor examination [1,2,3,4,5], more attention has been paid to their exploration in respect of the design and development of medicines [6,7,8,9,10], food quality classification [11,12,13,14], and disease diagnosis [15]. Moreover, a novel nanowire device S3, similar to a portable electronic nose, was successfully designed and served as an effective means to determine the authenticity of the grated Parmigiano Reggiano cheese [16]. At the same time, e-noses and e-tongues are gradually being employed in TCM research for both odor or taste determination [17] and quality evaluation [18,19]. Multi-sensor data fusion based on e-tongues and e-noses has the advantages of being simple and fast, and has been applied to food shelf-life assessment [20], beer flavor classification [21], qualitative and quantitative assessment of tea quality [22], etc., but few reports have been published about their application for TCMs. Hong Men [21] established three classification models based on extreme learning machine (ELM), random forest (RF), and support vector machine (SVM), to evaluate the efficiency of the feature-mining method. Comparable prediction accuracies of 94.44–98.33% were achieved, indicating an excellent classification performance of the three models. Cole [23] reported the combination of an e-nose and e-tongue for the flavor analysis of liquids. Only a combination of the two systems was able to achieve 100% discrimination among all the different liquids. Similarly, Banerjeeroy [24] combined measured data from an individual e-nose and e-tongue for improving the estimation of black tea quality. When compared to separate systems, combined systems improved both clustering and classification rates. Tian [25] studied the integration of an e-nose and e-tongue for adulteration detection of minced mutton, and low classification error (0–1.67%) evaluated by canonical discriminant analysis (CDA) and Bayesian discriminant analysis (BAD) was obtained by the combined system, suggesting a superior discrimination capability to that obtained by the e-nose (0.83–10.83%) and e-tongue (0–2.5%) separately. It was demonstrated that multi-sensor data fusion in food classification and quality assessment was much closer to human perception and significantly improved the accuracy of recognition. However, data fusion also results in redundant information, which could lead to unfavorable final classification and prediction or increase the complexity of the model prediction. Many pattern recognition methods such as principal component analysis (PCA), hierarchical cluster analysis (HCA), and partial least-squares discriminant analysis (PLS-DA) have been conveniently applied for data processing and simplification. In this paper, a multiple data fusion method based on e-nose measurements, e-tongue measurements, and chemical analysis coupled with chemometrics was developed for discrimination of the two botanical origins of Magnoliae Officinalis Cortex.

Magnoliae Officinalis Cortex (Houpo in Chinese, MOC) is widely distributed in the Hubei, Sichuan, Hunan, and Guangxi provinces of China and is sporadically distributed in East Asia. *Magnolia officinalis* Rehd. et Wils. (JYHP) and its variety *M. officinalis* Rehd. et Wils. var. *biloba* Rehd. et Wils. (AYHP) are included in the Chinese Pharmacopoeia as genuine herbs. The highly aromatic bark is stripped from their stems, branches, and roots and is used for the treatment of abdominal distention and pain, dyspepsia, relieving distension and asthmatic cough, etc. [26]. Phytochemical studies have led to the isolation of several active compounds including lignans [27,28,29,30], phenolic glycosides [31,32], phenylethanoid glycosides [33,34], and alkaloids [35,36,37]. Several studies have conducted useful explorations on the origin division, special processing, and quality evaluation of MOC using electronic nose, electronic tongue, and chemical composition analysis. Li [38] developed different discriminant models for differentiating the origins of MOC using a combination of an electronic nose and colorimeter. It was indicated that the random forest classifier combined with the tenfold cross-validation exhibited the highest accuracy, providing a promising method for quantitatively evaluating the quality of MOC. Zhao [39] reported methods for discriminating sweating and non-sweating processed MOC samples using HFLC–Qtof/MS and GC–MS/MS. A previous study also reported some differences in the pharmacological activities of the two origins of magnolia-based medicinal materials in the gastrointestinal tract [40]. Medicinal materials originating from AYHP were found to be superior to those from AYHP, which was in agreement with the traditional experience. JYHP and AYHP are easily distinguished at a botanical level by the tip of the leaf blade; however, they are extremely difficult to distinguish only by the appearance trait of their medicinal bark, especially after processing into medicinal materials. He et al. [41] used AFLP molecular markers to distinguish the two origins, but no significant differences between the two origins were observed. Thus, there has not been a suitable approach to effectively differentiate the two origins. Therefore, the overarching objectives of the current study were to use a combination of e-nose measurements, e-tongue measurements, and chemical analysis to successfully distinguish the two origins. However, the utilization of any technology can only reflect one aspect for the sample, which may cause inaccuracy of the classification. Thus, multiple data fusion coupled with different discriminant models, which may provide a higher accuracy, are additionally considered.

## 2. Materials and Methods

### 2.1. Materials

#### 2.1.1. Experimental Samples

Thirty samples originating from JYHP were collected from Enshi City of HuBei province and 30 originating from AYHP were from Yongzhou City of HuNan province. All the samples, derived from the diameter at breast height position to avoid errors caused by different parts, were authenticated by the authors and subjected to shade drying after collection. Samples were deposited at the Institute for Control of Chinese Traditional Medicine and Ethnic Medicine, National Institutes for Food and Drug Control, Beijing, China. Samples used for e-nose, e-tongue, and chemical data acquisition were pulverized to powder by passing through a no. 50 sieve (300 µm).

#### 2.1.2. Chemicals and Reagents

Methanol used for crude sample preparation was of analytical grade and was supplied by Beijing Chemical Factory (Beijing, China). HPLC-grade acetonitrile purchased from Merck KGaA (Darmstadt, Germany) was used to prepare the mobile phase. Water was purified by a Milli-Q water purification system (Millipore, Milford, MA, USA). All solvents were degassed by ultrasonication and an online degassing system. Analytical reagents including potassium chloride, potassium hydroxide, tartaric acid, absolute alcohol, and hydrochloric acid (35–37%, *v*/*v*) were all acquired from Sinopharm Chemical Reagent Co., Ltd. (Shanghai, China).

### 2.2. Method

#### 2.2.1. Preparation of Standard and Sample Solutions

Eight reference compounds of syringin, magnocurarine, magnoflorine, magnoloside B, magnoloside A, honokiol, magnolol, and β-eudesmol with batch Nos. Y28M9H57297, M29J8S40878, R21M9F61834, X16S8L44141, KS0912CB14, T28O6B5149, and P24O8F46474, respectively, were purchased from Shanghai Yuanye Bio-Technology Co., Ltd. (Shanghai, China). Piperitylmagnolol with a purity of more than 98.0% by normalization of the peak areas was isolated from the stem bark. Standard stock solutions of eight references were prepared with methanol at 37.40, 29.24, 30.72, 61.92, 88.32, 49.16, 45.60, and 29.16 mg/L concentrations and stored at 4 °C. The sample used for UHPLC and GC analysis was pulverized to powder with 50 mesh, and 0.3 g of powder was accurately weighed into 50 flasks with 25 mL of methanol. Following cold soaking for 24 h at room temperature, the sample mixture was weighed again, and any mass lost was made up with methanol. Subsequently, the sample solution was filtered through a 0.22 μm microporous membrane, and 1.0 µL of filtrate was subjected to chemical analysis.

A reference solution of 30 mM potassium chloride in 0.3 mM tartaric acid was used to simulate artificial salivain in the e-tongue assessment. A 30% ethanol aqueous solution containing 100 mM potassium chloride and 10 mM potassium hydroxide was employed as the cleaning solution for positively charged artificial lipid membrane sensor probes, while a 30% ethanol aqueous solution containing 100 mM hydrochloric acid was used for negative sensor probes.

#### 2.2.2. Data Acquisition for E-Nose Measurements, E-Tongue Measurements, and Chemical Analysis

##### E-Nose Data Acquisition

Characteristic odors of all the samples were provided with a portable electronic nose system PEN3 (Schwerin, Germany). The system consists of a gas collection unit, an air purification system, and a gas detection system, coupled with 10 metal oxide semiconductor (MOS) sensors. Different responses in the sensors provide a signal pattern characteristic of different volatile components as listed in Table 1 [42].

A sensor check was carried out to ensure that the sensors were working in the correct voltage range before measurement. An amount of 1 g of sample powder was placed in a 50 mL sampling chamber and capped with a PTFE septum. The headspace equilibrium of each sample was achieved by incubating at 30 °C for 60 min. To normalize the sensor signal, the gas chamber was first cleaned with gas filtered by active charcoal. Meanwhile, after each injection and data collection, the sensor self-cleaning time was at least 120 s to re-establish a stable instrument baseline. Sampling time was set at 1 s/group with a constant flow rate of 400 mL/min and a time of 120 s. Three duplicated cases were measured, and the data of 90 s were selected for the statistical analysis to ensure that the electronic nose reached the adsorption equilibrium.

##### E-Tongue Data Acquisition

E-tongue data acquisition was performed using the taste sensing system SA402B (Insent Inc., Atsugi-shi, Japan). Six sensors with differently composed lipid membranes and three corresponding reference electrodes were used. In the beginning, a sensor check was necessarily accomplished to ensure all the sensors were working in the correct voltage range before every measurement. Every sample measurement started with a cleaning procedure: the positive and negative cleaning solutions were applied for cleaning the sensors for 90 s, and then the sensors were cleaned with a standard solution for 120 s. The stability of the lipid membrane potential was monitored after cleaning by measuring the potential of the reference solution (Vr). The sample solution was tested for 30 s when the sensor response was steady (deviation less than 0.5 mV) during the measurement (Vs). The differential between Vr and Vs produced the sensor output for taste (R). A brief cleaning operation was performed to evaluate the samples’ aftertaste, which was referred to as CPA (change in membrane potential due to adsorption), by measuring the membrane potential again in standard solution [18]. Taste signals for each sample were measured four times, and the last three results were commonly used to ensure data stability.

Five taste sensors including AAE (umami), CT0 (saltiness), CA0 (sourness), C00 (bitterness), and AE1 (astringency) were performed to obtain the taste information. Three aftertastes of umami, bitterness, and astringency were also recorded. All recorded data were based on the absolute output value of artificial saliva (reference solution) as the standard. As a result of the limited amount of acid and salt in the reference solution, the tasteless points of sourness and saltiness were −13 and −6, respectively, while the output value of other indicators′ tasteless points was 0. In other words, when the taste value of the sample was lower than the tasteless point, it meant that the sample had no such taste. In our study, the taste items greater than the tasteless point were selected as the evaluation objects.

##### Multi-Component Quantitative Analysis

Quantitative analysis of eight active components comprising syringin, magnocurarine, magnoloside A, magnoflorine, magnoloside B, honokiol, magnolol, and piperitylmagnolol was performed using an ultrahigh-performance liquid chromatography (UHPLC) system. According to our previous study [43], chromatographic analysis was conducted on a suitable guard column (Waters Acquity UPLC BEH-C_l8_ 50 mm × 2.1 mm, 1.7 μm), and separation was performed using gradient elution. All nonvolatile ingredients were identified by comparison with the ultraviolet spectrum or retention time of the reference compounds. The contents of β-eudesmol were separately determined by a GC system equipped with an FID. An Agilent HP-5 quartz capillary column of 30 m length, 0.25 mm internal diameter, and 0.25 μm film thickness (J&W Scientific Inc., Folsom, CA, USA) was used for chromatographic separation. The FID temperature was set at 250 °C, and nitrogen was chosen as the carrier gas with a constant flow of 1 mL·min^−1^. The injection volume was 1.0 μL, and the split ratio was 15:1. The oven time–temperature program was as follows: initial temperature of 80 °C held for 2 min, increased to 250 °C at 8 °C/min, held for 7 min.

#### 2.2.3. Chemometric Analysis and Data Fusion

Chemometric analysis including PCA, HCA, PLS-DA, LDA, and QDA was conducted using OriginPro 2021b software (OriginLab Corporation, One Roundhouse Plaza, Northampton, USA). As an unsupervised pattern identification method, PCA can convert the data into a new coordinate system and produce numerous new synthetic variables. It displays linear combinations of them while also capturing the majority of the original data’s properties. The score value plots for the first two or three PCs (PC1, PC2, and PC3) are frequently used to visualize the sample characteristics. The quality of the fitting model may be explained by the modeling parameters, *R^2^*X and *Q^2^* values in PCA. In contrast to PCA analysis, which introduces a lot of unnecessary noise into the regression modeling process, partial least-squares discrimination analysis (PLS-DA) is a supervised discriminant analysis statistical method that employs information synthesis and screening techniques in the regression modeling process. More importantly, the variable importance for projection (VIP) can be employed to screen the potentially important variables with the principle of value >1.0. According to the VIP plot, crucial variables are primarily selected and are significantly responsible for the discrimination.

Classification models including LDA and QDA were explored in this study. Three validation approaches comprising resubstitution, cross-validation, and a sample dichotomy strategy were adopted to help evaluate model reliability. For resubstitution, all of the data were used for establishing the model, which then classified this same dataset. For cross-validation, the leave-one-out technique was applied; that is, the models were trained using all of the data minus one sample, which was then brought into the model for classification [44]. This procedure was repeated for each sample, and the number of correctly classified samples was presented at the end. For the sample dichotomy strategy, two-part samples were randomly separated into training sets to generate a predictive model, and the remaining samples were selected for the prediction set to gauge classification performance.

For the fusion of e-nose or e-tongue data, Di Rosa reported three methods of abstraction [45]. Simply concatenating the original data from the multiple sources for model construction is considered low-level fusion. Mid-level fusion involves the fusion of extracted features. Feature extraction with the advantage of eliminating the redundant information and multiple collinearity problem is performed for each data source. Separate models built from each data source are combined to give high-level fusion. In this paper, low-level and mid-level fusion methods were preliminary investigated for the fusion of e-nose, e-tongue, and chemical analysis data.

## 3. Results and Discussion

### 3.1. Results of E-Nose Measurements, E-Tongue Measurements, and Chemical Analysis

Odor information for 30 samples originating from JYHP and 30 from AYHP was collected at the adsorption equilibrium point time (90 s). Each sample was measured three times to ensure the stability of the odor data. The e-nose detection data for all the samples containing characteristic values of 10 sensors are shown in Appendix A. As shown in Figure 1a, among the 10 odor sensors, W1W, W2W, and W5S sensors presented the strongest responses; the responses of W1W, W2W, W5S, W1S, and W2S of the JYHP group were significantly higher than those of the AYHP group. Conversely, the responses of W1C, W3C, and W5C presented the opposite trend. For the e-nose sensor response, we performed Pearson correlation with the content of the volatile β-eudesmol, but no correlation was found. It can be inferred that the e-nose sensor response represents the overall performance of multiple volatile components. Jing et al. [46] realized the comparative analysis of volatiles in the seeds of AYHP and JYHP, and the results revealed that the volatile profiles were different between AYHP and JYHP. The study provides evidence that volatiles traits in the seeds of AYHP and JYHP, in accordance with the morphological properties of their leaves, are controlled by genetic and environmental factors. According to this finding, the response values of e-nose sensors, connected to the volatile component composition, should be different between the two origins.

The measurement results of bitterness, astringency, umami and richness, aftertaste B, and aftertaste A are concluded in Appendix A. Figure 1b provides results showing that samples from different origin groups may present different taste information. All samples displayed the highest response values for bitterness, which agreed well with the taste description of the MOC features in the ChP. The experimental data generated for astringency, umami, aftertaste B, and aftertaste A were subjected to one-way analysis of variance (ANOVA); differences were considered significant at *p* < 0.05 between the two origin groups. When the bitterness and richness data were used for Welch’s *t*-test due to heterogeneity of variance, the same significant difference (*p* < 0.05) was also observed, indicating that the taste information between the two origins was inconsistent.

The acquired results of the chemical analysis, as illustrated in Appendix A, revealed that the nine analyzed compounds included in the different origin types varied significantly. Total contents of honokiol and magnolol (THM), as the quality control markers of MOC, with a minimum of 2.0% of the total amount in ChP, were most abundant. These were in line with a previous phytochemical study showing that honokiol and magnolol are considered the major bioactive constituents in these herbs [28]. Welch’s *t*-test showed that THM between the AYHP and JYHP groups was obviously different, with *p* < 0.05. The JYHP group had a higher THM content level compared with AYHP, suggesting its superior quality to AYHP. This was confirmed by previous findings, showing that the effect of improving gastrointestinal motility of JYHP is superior to that of AYHP. Another phenolic component, named piperitylmagnolol, was simultaneously detected, and the JYHP group presented considerably higher contents compared with the AYHP group. Hence, these three phenolic components seem to play an important role in the identification of AYHP. Phenylethanoid glycosides, widely distributed in MOC, have aroused great interest in the last 10 years due to their pharmacological activity [33]. Magnolosides A and B, as the typical phenylethanoid glycosides, are commonly used as markers for evaluating the quality of commercial samples [47]. Quantitative profiles of magnolosides have also been designed for the discrimination of MOC from different geographical regions [48]. The distributions of magnolosides A and B in this study, as clearly shown in Figure 1c, exhibited a cross-current between the two groups. All of these characteristics are possibly essential considerations in differentiating the two origins. The proportionate features among homologous substances for determining the source, origin, and quality of Chinese herbs were previously investigated [49]. Thus, the proportionate features of honokiol and magnolol (PHM), along with magnolosides B and A (PBA), were included in the multivariate dataset to discriminate the two origins.

### 3.2. Comparison Results of E-Nose Measurements, E-Tongue Measurements, and Chemical Anylsis on Discrimination of the Two Origins

As an unsupervised data dimensionality reduction approach, principal component analysis (PCA) does not consider sample group information but is frequently beneficial for data exploration, occasionally showing visual grouping. Figure 2a shows the PCA biplot for the e-nose; the variance explained by the first component (PC1) was 71.7%, while the second component (PC2) explained 14.4%, accounting for a total 86.1% of the variability. W2S, W1S, W5S, W1C, W3C, and W5C were sensitive to the variables significantly contributing to PC1 and exhibiting similar behavior. W2S, W1S, and W5S, which were sensitive to broad range alcohols, short-chain alkanes, and hydrocarbons, respectively, were negatively related to W1C, W3C, and W5C. The score plot shows a moderate separation between the two origins. With respect to the PCA results for the e-tongue measurements, aftertastes A and B and richness were the variables crucially contributing to PC1, which explained 58.8% of the variance of the results. PC2, including the variables other than aftertaste A and richness, explained approximately 20.5% of the data variance. In addition, only the bitterness exhibited an eigenvalue with 0.577 > 0.5, revealing a significant contribution to PC2. It was implied that a poor correlation between the taste sensors’ responses and moderate predictive ability of the PCA model may be presented.

The biplot of the chemical analysis shows that PHM and PBA mainly contributed to PC1, which explained approximately 53.4% of the data variance (i.e., 53.4% and 14.3% for PC1 and PC2, respectively). Two PCs could explain 67.7% of the variability, suggesting that the extracted PCs had general explanatory power for the original variables. The score plot seemed to afford good separation of the samples except in the two misclassified cases. The variables syringin, magnocurarine, magnoflorine, and magnoloside A were grouped into the same quadrant due to their high content distribution in AYHP compared with JYHP. PC1 comprised PHM, PBA, piperitylmagnolol, THM, honokiol, magnolol, and magnoloside B. This seems to be related to their high score in JYHP in contrast with AYHP.

Hierarchical cluster analysis (HCA) was exercised for better visualization of the differences between the two botanical origins. The Euclidean distance was applied to explore the similarity. Figure 3a shows that the overwhelming majority of those samples from different origins could be correctly classified according to the e-nose data. However, samples including S2, S5, S10, S4, and S8 belonging to JYHP were clustered together and grouped into adjacent clusters of the AYHP. This could be attributed to the W5S sensor response of these samples being significantly lower than that of samples belonging to JYHP. The HCA dendrogram for the e-tongue (Figure 3b) demonstrated a mediocre separation of the two origins, where seven samples from JYHP were misclassified to the AYHP cluster. No obvious abnormality was found in the electronic tongue measurement values of these samples. The HCA dendrogram (Figure 3c) from the chemical analysis shows that samples from different origins were employed to respective clusters except S17, S27, and S60. The misclassification of S17 and S27 may be due to their low magnoloside B contents. S60, belonging to AYHP, was clustered to JYHP and associated with its high honokiol content compared with the other cases of JYHP. Overall, all three approaches could discriminate the two origins with a percentage of misclassified samples. The HCA for the chemical analysis appeared to perform somewhat better than that for the e-nose and e-tongue. PCA and HCA were classified according to the similarity of sample features and affinity relationship, and it was inevitable that there would be misclassification. Therefore, supervised discriminant analysis may provide a more effective classification.

Linear discriminant analysis (LDA) and quadratic discriminant analysis (QDA) were investigated in this pattern classification with up to three evaluation strategies considered. The results of classification modeling and evaluation approaches with numbers and percentages of misclassifications are summarized in Table 2. The data indicate that the e-nose technique affords good classification rates with a percentage of misclassifications in the range 0–1.67%, and better discrimination results were obtained by QDA. In particular, no misclassified cases were observed with sample dichotomy strategy validation, demonstrating an excellent dependability of LDA and QDA based on the e-nose. For the e-tongue, the percentage of misclassifications was in the range 0–3.33%, and the QDA displayed the best discrimination results with resubstitution validation. However, the chemical analysis with respect to both LDA and QDA exhibited poor success rates. This deserves our attention, as errors in the classification of the QDA model with cross-validations were as high as 10.0%. On the basis of these results, we guardedly consider that an unknown sample from a different origin would in fact be classified correctly by such a model. To sum up, the result for the e-nose with the LDA model seems to be suitable for the discrimination of the two origins. However, the e-nose and e-tongue detection, and the chemical analysis show only one aspect of sample information. The evaluation of traditional Chinese medicine has always involved a combination of odor, taste, and appearance, or chemical properties. Thus, the interaction between each sensor’s response and the combination of all the detected information should be given careful consideration to obtain better identification results.

### 3.3. Extraction of Feature Variables Based on PCA and PLS-VIP

Feature extraction by PCA from the three respective data sources was developed as described in Section 2.2. In order to retain the most information from the original data, 95.0% cumulative calibrated variance was acquired. Thus, the first four PCs from the e-nose, four PCs from the e-tongue, and seven PCs from the chemical profiles, accounting for 97.54%, 95.96%, and 96.93% cumulative variance, respectively, were employed to create a new combined dataset. The new dataset with midlevel fusion, comprising a total of 15 PCs, was used to obtain a better discrimination result. Furthermore, the eigenvalue represents the contribution of the corresponding eigenvector after the matrix was orthogonalized. Eigenvalues greater than 1.0 are frequently used as the basic principal component extraction criterion. Thus, two PCs from the e-nose, two PCs from the e-tongue, and three PCs from the chemical profiles were eventually selected, explaining 86.14%, 79.30%, and 78.57% of the total cumulative variance, respectively. A new dataset was also established with these obtained PCs for mild data fusion, and comparison results were finally obtained.

A PLS-VIP method for detecting crucial variables contributing significantly to the origin classification was simultaneously proposed. The obtained data were pretreated by mean centering and scaling to unit variance. The classification ability and prediction effect of the PLS model were described as variations in the response Y (class) *R*^2^Y and *Q*^2^. The results of *R*^2^Y and *Q*^2^ were all greater than 0.5, indicating that the PLS model established from the three original data sources presented a satisfactory classification capacity. In addition, a chance permutation test (*n* = 200) was carried out to validate the goodness of fit and the predictability of the model. The intercept values of *R^2^* and *Q^2^* were less than 0.3 and 0.05, respectively, suggesting a desirable significance and no overfitting, with high predictive value of the three models. Potential markers for the separation of different origins were obtained by filtering VIP values exceeding 1.0.

As illustrated in Figure 4, W1C, W2W, W3C, W5C, W1S, and W2S from the e-nose possessed VIP scores greater than 1.0, suggesting that these variables play a key role in origin discrimination. As to the e-tongue, the sensors for umami and astringency presented a VIP value >1.0, suggesting their importance for distinguishing the two origins. Astringency is a complicated phenomenon in which some food (unripe fruits, wines, and teas) or medication substances (polyphenols) induce the mouth epithelium to pucker or dry [50]. A previous report showed that the amount of galloyl rings present in the polyphenol chains affects the degree of astringency [51]. However, up to now, studies regarding astringency have faced challenges. Astringency may be associated with stimulation, which leads to enzymes, mucins, and PRPs being precipitated and results in the perception of astringency. According to the traditional medicine experience, MOC contains a certain irritant to the throat and is difficult to process, which could explain its astringency. It was observed that the astringency taste value of AYHP was lower than that of JYHP; conversely, the AYHP types showed greater intensity of the umami taste. These findings indicate that each origin type has distinct taste characteristics. Thus, the umami and astringency responses were selected as the crucial variables for the e-tongue. Five markers comprising honokiol, piperitylmagnolol, PHM, PBA, and THM were obtained from the chemical analysis of the PLS-DA model. Figure 1c shows that the selected markers were mainly at greater levels in the JYHP group than in the AYHP group. This might be explained by the fact that JYHP samples have always been recognized as the traditional authentic medicinal material with better quality.

### 3.4. Multi-Source Data Fusion and Establishment of Discriminative Model

Although the e-nose, e-tongue, and chemical profiles could partly discriminate samples from the two origins, there were still a few misclassified cases according to the HCA dendrograms (Figure 3). With the exception of the LDA from the e-nose data, none of the discriminant analyses achieved 100% discrimination. Consequently, the data from the three sources were combined using low-level and mid-level fusion methods. Firstly, the original data from the three sources including both useful and redundant information were combined directly to form a new dataset; the LDA and QDA were obtained after the data normalization process. The results show that 100% of the original grouped cases were correctly classified by LDA, but there was a 56.67% misclassified rate generated by QDA with cross-validation (Table 3). We further tried to process the combined original data using PCA. The first 11 principal components, totally explaining 95.10% cumulative variance, were selected. The calculated PC scores were used as independent metrics to replace the original variable for the LDA and QDA. The LDA result was in agreement with that acquired using raw data, and the error probability of QDA significantly decreased after data dimensionality reduction for cross-validation. In order to present more information, 11 principal components were selected as statistical variables, which resulted in successful classifications. However, if the PCs were selected according to the principle of eigenvalues >1.0, seven PCs were obtained, explaining 87.43% of the total varieties. Therefore, these seven PCs, as variables that did not reduce the discriminant rate, perhaps yielded a more realistic classification, but the contribution of such reduced PCs to the model is still uncertain.

For the mid-level fusion, the PCs extracted from the three original datasets and the extracted variables based on PLS-VIP were combined and explored with respect of LDA and QDA. As a result, LDA seemed to work consistently better with no misclassified cases as validated by the different methods. QDA presented a classification error of 1.67–3.33% for cross-validation due to its possible overfitting. It can be seen that the data fusion processes showed good results with regard to the classification of the two origins by resubstitution and sample dichotomy strategy validation. The 15 PCs extracted from the three subsets were obtained according to the principle of containing more than 95.0% of the total cumulative variance. However, the quantity of the extracted PCs was decreased to seven according to the principle of an eigenvalue greater than 1.0. The LDA result was unaffected by the reduction in these PCs, but QDA showed a decrease in the misclassification rate from 3.33% to 1.67% for cross-validation. Similarly, 1.67% of misclassification rates also appeared in combination with extracted variables based on a VIP >1.0. Therefore, according to the results of the three mid-level fusion methods, the LDA delivered the optimal results with 100% discrimination for all validations, indicating that this model has a high prediction accuracy.

The discrimination capability of the LDA model was significant according to the eigenvalue and the Wilks lambda test. The corresponding eigenvalue of the discriminant function explained all of the sample information with a l00% variance ratio. The *p*-value of the correlation coefficient was less than 0.05. All of these results reveal that the LDA model built from the three mid-level fusion methods exhibited excellent performance and was suitable for distinguishing the two origins through significance testing. PLS is a supervised method that incorporates the ideas of principal component analysis and canonical correlation analysis. The purpose of using PLS to reduce dimensionality is to make the extracted feature variables not only summarize the information from the original variables well, but also have strong explanatory power for the dependent variables. Hence, a combination of extracted variables from PLS-VIP can be observed more intuitively and may be convenient for practical use. In short, the e-nose and e-tongue sensor arrays and chemical analysis, coupled with mid-level data fusion, could successfully be applied in the discrimination of the two botanical origins of MOC.

## 4. Conclusions

In this study, we initially reported the discrimination of MOC from two botanical origins (*Magnolia officinalis* Rehd. et Wils. and *Magnolia officinalis* Rehd. et Wils. var. *biloba* Rehd. et Wils) based on the multiple-source fusion of e-nose, e-tongue, and chemical analysis data. Feature mining was conducted using PCA; 15 PCs with the principle of the total cumulative variance contribution rate exceeding 95.0% and seven PCs with the principle of eigenvalues >1.0 were extracted. In addition, 13 variables were mined according to the principle of PLS-VIP scores >1.0. Multiple-source data fusion based on low-level and mid-level data was subsequently conducted. The LDA and QDA models, with up to three evaluation strategies, were investigated to acquire the efficiency of the feature-mining method. As a result, successful predictability and validation of the LDA model with 100% correct classification were observed, indicating that the combination of these three techniques could effectively enhance the discriminate performance. As our results revealed, mid-level data fusion, especially for a combination of extracted variables based on PLS-VIP, was concise and feasible. However, it is worth noting that MOC also contains volatile components other than β-eudesmol; hence, the relationship between their contents and the response values of the e-tongue and e-nose is ambiguous. Their contribution to this origin discrimination is still unknown and worthy of a deep investigation. Other classification models including logistic regression analysis, backpropagation neural network (BPNN), random forest (RF), and support vector machine (SVM) can also be applied upon increasing the number of samples. Thus, we explored a completely simple and practicable approach to identify the origin of MOC on the basis of e-nose, e-tongue, and chemical analysis, as well as data fusion technology. This study could pave a way for its application in further quality evaluations.

## Figures and Tables

**Figure 1 molecules-27-03892-f001:**
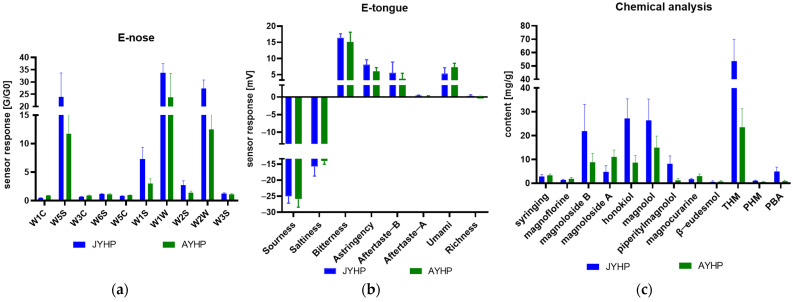
Statistical results of e-nose (**a**), e-tongue (**b**), and chemical analysis (**c**) of the primitive samples from two origins (JYHP and AYHP). G: conductivity of the sensor after contact with the sample gas; G0: conductivity of the sensor after cleaning with standard activated carbon filter gas.

**Figure 2 molecules-27-03892-f002:**
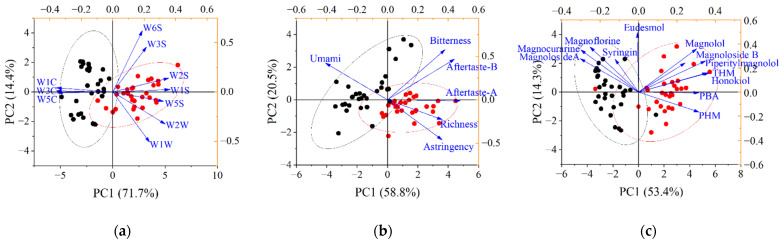
PCA analysis biplot: loading plot of the first two principal components with dataset from the e-nose (**a**), e-tongue (**b**), and chemical analysis (**c**); sample score plot where AYHP and JYHP samples are marked in black and red, respectively.

**Figure 3 molecules-27-03892-f003:**
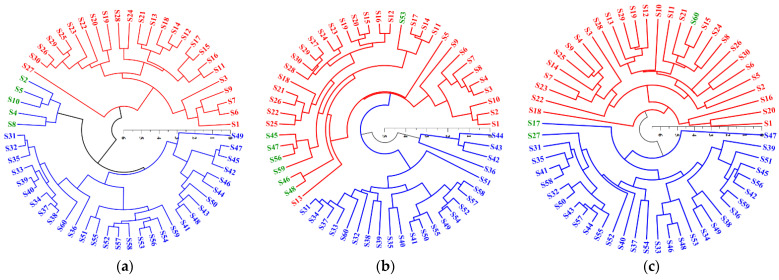
Established HCA dendrograms of two origin samples with data acquired from the e-nose (**a**), e-tongue (**b**), and chemical analysis (**c**).

**Figure 4 molecules-27-03892-f004:**
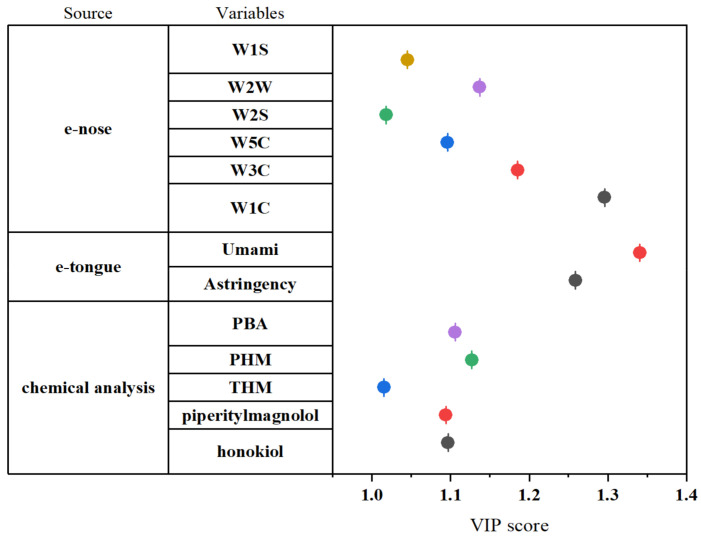
Selected variables with VIP score >1.0 from e-nose, e-tongue, and chemical analysis.

**Table 1 molecules-27-03892-t001:** Sensor description of e-nose.

No	Sensor Name	Performance Description	Sensitive Substances and Threshold Values (mL·m^−3^)
1	W1C	Aromatic	Toluene, 10
2	W5S	Hydrocarbon	Nitrogen dioxide, 1
3	W3C	Aromatic	Benzene, 10
4	W6S	Hydrogen	Hydrogen, 100
5	W5C	Aromatic and aliphatic	Propane, 1
6	W1S	Broad range and methane	Methane, 100
7	W1W	Sulfur organic	Hydrogen sulfide, 1
8	W2S	Broad range alcohol	Nitric oxide, 100
9	W2W	Sulfur and chlorinate	Hydrogen sulfide, 1
10	W3S	Methane and aliphatic	Methane, 10

**Table 2 molecules-27-03892-t002:** Comparison results of LDA and QDA based on e-nose, e-tongue, and chemical analysis using different validation methods.

Model	Data Source	Resubstitution		Cross-Validation		Sample Dichotomy Strategy	
		NM ^1^	PM ^2^	NM	PM	NM	PM
LDA	E-nose	1	1.67%	1	1.67%	0	0
	E-tongue	1	1.67%	2	3.33%	1	1.67%
	Chemical analysis	2	3.33%	3	5.00%	T:2	5.00%
QDA	E-nose	0	0	0	0	0	0
	E-tongue	0	0	1	1.67%	1	1.67%
	Chemical analysis	2	3.33%	6	10.00%	T ^3^:1 P ^4^:2	2.50% 5.00%

^1^ NM, number of misclassified cases; ^2^ PM, percentage of misclassified cases; ^3^ T, training set; ^4^ P, prediction set.

**Table 3 molecules-27-03892-t003:** Percentage of samples misclassified by LDA and QDA using different data fusion methods and validations.

Fusion	Data Source	Model	Resubstitution		Cross-Validation		Sample Dichotomy Strategy	
			NM ^1^	PM ^2^	NM	PM	NM	PM
Low-level fusion	Original data with normalization	LDA	0	0	0	0	NA ^3^	NA
		QDA	0	0	34	56.67%	NA	NA
	Original data with PCA	LDA	0	0	0	0	0	0
		QDA	0	0	1	1.67%	0	0
Mid-level fusion	Combination of 15 extracted PCs	LDA	0	0	0	0	0	0
		QDA	0	0	2	3.33%	0	0
	Combination of 7 extracted PCs	LDA	0	0	0	0	0	0
		QDA	0	0	1	1.67%	0	0
	Combination of extracted variables	LDA	0	0	0	0	0	0
		QDA	0	0	1	1.67%	0	0

^1^ NM, number of misclassified cases; ^2^ PM, percentage of misclassified cases; ^3^ NA, not applicable.

## Data Availability

Not applicable.

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
