# Peer review of "Application of Multiple-Source Data Fusion for the Discrimination of Two Botanical Origins of Magnolia Officinalis Cortex Based on E-Nose Measurements, E-Tongue Measurements, and Chemical Analysis"

_molecules, 2022, doi:10.3390/molecules27123892_

Round 1

Reviewer 1 Report

Comments for the authors:

  • The Introduction must contain more recent references and research findings on the topic and present a comparison (discrimination ability) of this study with the cited works.
  • The font size of Figure 2 must be increased for a better readability.
  • It is advised to present the sensors used in e-nose and e-tongue in a tabular format separately with the sensors characteristics, major VOCs detected, working temperature and response time.
  • The major disadvantages and drawbacks of the proposed approach of classification must be highlighted.
  • The English and grammatical errors, typos must be corrected throughout the manuscript.
  • The authors are advised to cite the following papers to enhance its quality:
  • Natural Product Communications Vol. 13 (8) 2018, 987-991.
  • https://doi.org/10.1016/j.jtcms.2020.03.004
  • https://doi.org/10.1007/978-3-319-47322-2_11

Author Response

The response to the reviewer's comments, please see the attachment.

Reviewer 2 Report

In this study,  discrimination of MOC from two botanical origins based on the multiple-source fusion of e-nose, e-tongue, and chemical analysis data. As an agricultural and botanic application of e-nose and e-tongue, the results are interesting. 

The quality of presentation is very low. The methods are not adequately described. Why the results were presented and discussed before the materials and methods are clearly explained?

Information about the e-nose and e-toungues are not described sufficiently. The sensor properties and response characteristic tests agains well known volatile VOC gasses like ethanol, methanol etc.  have been not given. How can you measure if the sensor responding adequately or not? 

Figure 1. is not clear. How can we compare them without measure units on the plot axis?

The interpretation of the very long Table 1, Table 2 and Table 3 are not clear. Again no unit was given in the table. What are you measure? with quantity?

No meaning to give so long tables instead of plotting to see the changes on the parameters. What is the relation between Figure 1 and the 3 tables are not clearly described.  Presenting all the sensor responces as a table up to s60 not an effective way to convince the readers. They should be given in a supporting material part with a reference. 

Table 4. should be controlled carefully. "cross-validated - Number of misclassified" is listed as percentage. The same mistake for "sample dichotomy strategy".

The overall manuscript should be carefully reorganized and presented with the necessary informations discussed in details. 

Author Response

response to the reviewer's comments, please see the attachment.

Round 2

Reviewer 2 Report

The authors have modified most of the reviewer suggestions. It can be accepted for publication as it is.

This manuscript is a resubmission of an earlier submission. The following is a list of the peer review reports and author responses from that submission.